# Osteoarthritis-Related Inflammation Blocks TGF-β’s Protective Effect on Chondrocyte Hypertrophy via (de)Phosphorylation of the SMAD2/3 Linker Region

**DOI:** 10.3390/ijms22158124

**Published:** 2021-07-29

**Authors:** Nathalie Thielen, Margot Neefjes, Renske Wiegertjes, Guus van den Akker, Elly Vitters, Henk van Beuningen, Esmeralda Blaney Davidson, Marije Koenders, Peter van Lent, Fons van de Loo, Arjan van Caam, Peter van der Kraan

**Affiliations:** 1Department of Experimental Rheumatology, Radboud University Medical Center, 6500 MD Nijmegen, The Netherlands; Nathalie.Thielen@radboudumc.nl (N.T.); Margot.Neefjes@radboudumc.nl (M.N.); Renske.Wiegertjes@radboudumc.nl (R.W.); Elly.Vitters@radboudumc.nl (E.V.); Henk.vanBeuningen@radboudumc.nl (H.v.B.); Esmeralda.BlaneyDavidson@radboudumc.nl (E.B.D.); Marije.Koenders@radboudumc.nl (M.K.); Peter.vanLent@radboudumc.nl (P.v.L.); Fons.vandeLoo@radboudumc.nl (F.v.d.L.); Arjan.vanCaam@radboudumc.nl (A.v.C.); 2Department of Orthopedic Surgery, Maastricht University, 6200 MD Maastricht, The Netherlands; g.vandenakker@maastrichtuniversity.nl

**Keywords:** TGF-β, osteoarthritis, cartilage, SMAD2/3 signaling, linker modifications, inflammation

## Abstract

Osteoarthritis (OA) is a degenerative joint disease characterized by irreversible cartilage damage, inflammation and altered chondrocyte phenotype. Transforming growth factor-β (TGF-β) signaling via SMAD2/3 is crucial for blocking hypertrophy. The post-translational modifications of these SMAD proteins in the linker domain regulate their function and these can be triggered by inflammation through the activation of kinases or phosphatases. Therefore, we investigated if OA-related inflammation affects TGF-β signaling via SMAD2/3 linker-modifications in chondrocytes. We found that both Interleukin (IL)-1β and OA-synovium conditioned medium negated SMAD2/3 transcriptional activity in chondrocytes. This inhibition of TGF-β signaling was enhanced if SMAD3 could not be phosphorylated on Ser213 in the linker region and the inhibition by IL-1β was less if the SMAD3 linker could not be phosphorylated at Ser204. Our study shows evidence that inflammation inhibits SMAD2/3 signaling in chondrocytes via SMAD linker (de)-phosphorylation. The involvement of linker region modifications may represent a new therapeutic target for OA.

## 1. Introduction

Osteoarthritis (OA) is characterized by irreversible cartilage breakdown and regarded as a multifactorial disease in which inflammation is involved [1,2]. Synovitis is present in osteoarthritic joints and the production of inflammatory cytokines and chemokines is increased [3,4,5,6]. These pro-inflammatory cytokines, such as Interleukin (IL)-1β can have a direct (negative) effect on cartilage homeostasis [7,8], but can also modulate the transforming growth factor-β (TGF-β) signaling [9,10].

TGF-β is a crucial growth factor for articular cartilage maintenance [11]. Via intracellular activation of the transcription factors SMAD2 and SMAD3, TGF-β inhibits chondrocyte hypertrophy and MMP13 expression [12]. On the other hand, signaling via its alternative SMAD1/5/9 signaling route promotes these detrimental processes. A disturbed balance between the two SMAD signaling routes has been proposed as a cause for OA pathology [13,14]. TGF-β signaling disruption can occur at different stages in its signaling cascade. For instance, inflammatory mediators can regulate the TGF-β receptor expression and increase the expression of inhibitory SMAD7 [15]. Alternatively, inflammatory pathways can induce post-translational modification of the linker region of SMAD proteins to modulate their function [16,17]. This linker domain connects the N-terminal MH1 domain, which is important for DNA binding and nuclear transport, to their C-terminal MH2 domain, which is responsible for the SMAD receptor and SMAD–SMAD interactions and gene transcription activation [18,19,20,21]. Importantly, the linker region can be phosphorylated on specific serine and threonine residues, and this regulates nuclear entry, SMAD–protein interactions, and SMAD turnover, thereby greatly affecting SMAD function [16,22,23]. Still, the relative importance of these SMAD linker modifications in cartilage biology and OA pathogenesis has not been investigated and is poorly understood.

In this study we explored whether OA-related inflammation dysregulates TGF-β signaling in chondrocytes via inflammation-driven SMAD2/3 protein linker-modifications.

## 2. Results

### 2.1. IL-1β and OAS-cm Negate the Anti-Hypertrophic Function of TGF-β in Bovine Cartilage Explants

Hypertrophy-like changes in chondrocytes play a role in OA progression [24]. To study whether inflammation modulates such changes, a model for hypertrophy was set-up. For this, we cultured bovine cartilage tissue explants for 2 weeks ex vivo which, both with and without the addition of FCS in the culture medium, induced hypertrophy-like differentiation, as confirmed by a ~97-fold increase in *COL10A1* expression (2^6.6 ΔCt^, *p* < 0.0001) (Figure 1A). To demonstrate the anti-hypertrophic function of the TGF-β ex vivo, recombinant human (rh), TGF-β1 was added to culture medium (without FCS) every 3rd day. *COL10A1* expression was dose-dependently inhibited by TGF-β, with an EC_50_ of 0.1 ng/mL and 85% inhibition at 1.0 ng/mL TGF-β (2^5.6 ΔCt^, *p* = 0.0002) (Figure 1A). Co-incubation with the ALK-5 kinase activity inhibitor SB-505124 fully blocked TGF-β’s effect on *COL10A1* expression with a 74-fold difference compared with the vehicle (DMSO) (2^6.2 ΔCt^, *p* = 0.0021) (Figure 1B). To study the interaction between TGF-β and inflammatory mediators in this model of hypertrophy, explants were exposed to 0.1 ng/mL TGF-β combined with 0.1 ng/mL IL-1β or 0.5% OA synovium-conditioned medium (OAS-cm). Importantly, we first established that these concentrations did not modulate *COL10A1* expression themselves (Appendix A). Pre-incubation of explants for 1 h with IL-1β prior to the addition of TGF-β negated anti-hypertrophic TGF-β signaling with ~2.2 fold difference (2^1.2 ΔCt^, *p* = 0.0144) (Figure 1C). The addition of OAS-cm prior to TGF-β strikingly negated anti-hypertrophic TGF-β signaling with a 7.0-fold difference. (2^2.8 ΔCt^, *p* = 0.0113) (Figure 1D).

### 2.2. IL-1β and OAS-cm Inhibit TGF-β Transcriptional Activity in Different Chondrocyte Cell Lines

Hereafter, we used three different human chondrocyte-like cell lines (G6, H11, SW1353) to identify the cause of this interaction between functional TGF-β signaling and inflammatory mediators. We chose to use cell lines because it is difficult to efficiently genetically modify primary chondrocytes in explants culture. First, we established that a similar inhibitory effect occurs in these cell lines as in cartilage explants on TGF-β transcriptional activity. To perform this, we made use of a SBE-pNL1.2 luciferase reporter assay, which is SMAD2/3 and SMAD4-dependent (Appendix B). In all three cell lines, the luciferase signal was induced by TGF-β stimulation alone and this effect was significantly inhibited when pre-incubated for either 1 or 16 h with 0.1 ng/mL IL-1β or 0.5% OAS-cm (Figure 2A and Appendix B). This inhibition was further investigated in SW1353 cells (Figure 2B,C). Pre-incubation with 0.001 ng/mL IL-1β (area under the curve (AUC) = 86, *p* = 0.95) and 0.01 ng/mL IL-1β (AUC = 70, *p* = 0.12) did not inhibit TGF-β transcriptional activity (AUC = 92). However, 0.1 ng/mL IL-1β (AUC = 48, *p* = 0.0018), 1 ng/mL IL-1β (AUC = 40, *p* = 0.0004) and 10 ng/mL IL-1β (AUC = 33, *p* = 0.0002), pre-incubated for 1 h did significantly inhibit TGF-β transcriptional activity (Figure 2B). Pre-incubation with OAS-cm for 1 h inhibited TGF-β transcriptional activity (AUC = 85) from 35% inhibition with 1% OAS-cm (AUC = 54, *p* = 0.0063) up to 83% inhibition with 10% OAS-cm (AUC = 13, *p* < 0.0001) (Figure 2C). Combining these data supports the conclusion that OA-related inflammation has an inhibitory effect on TGF-β signaling in chondrocytes.

### 2.3. IL-1β and OAS-cm Do Not Inhibit C-Terminal Phosphorylation of SMAD2/3 and Do Not Regulate Receptor Level Expression

Upon TGF-β binding to the receptor, R-SMAD transcription factors become activated by phosphorylation of serine residues on their carboxy (C)-terminus, which causes them to form a complex with co-SMAD4, translocate to the nucleus and regulate gene transcription [25]. In search of an explanation for the strong inhibition of OA-related inflammatory factors on TGF-β signaling, we first investigated if IL-1β and OAS-cm influence C-terminal SMAD phosphorylation. As expected, TGF-β supplementation strongly increased pSMAD2/3C in both primary bovine chondrocytes and SW1353 cells, whereas IL-1β or OAS-cm did not (Figure 3A). However, pSMAD2/3C was not decreased by either 1, 6 or 24 h pre-incubation with IL-1β or OAS-cm (Figure 3A, upper panels), which excluded a direct effect on C-terminal SMAD phosphorylation. Another underlying cause for the disturbed TGF-β signaling could be a shifted balance from protective pSMAD2/3 to deleterious pSMAD1/5 [26,27]. However, in both primary chondrocytes and SW1353 cells, we observed that pSMAD1/5C was also not affected by 1, 6 or 24 h pre-incubation of IL-1β or OAS-cm (Figure 3A, middle panels). Together, these observations strongly indicate that the TGF-β-receptor complexes were unaffected. In support of this, we did not find changes in the receptor expression at the mRNA level. The stimulation of both primary chondrocytes and SW1353 cells with 0.1 ng/mL IL-1β or 5% OAS-cm for 1 h did not alter *TGFBR2* or *ALK5* receptor levels (Figure 3C), nor did stimulation for 6 h with IL-1β. Note that, in SW1353 6 h stimulation with 0.1 ng/mL, IL-1β did induce (and not reduce) *TGFBR2* (*p* = 0.0426) and did not change *ALK5* expression.

The duration and intensity of the SMAD depends on the abundance and availability of ligands and their inhibitors [28]. We also confirmed that the signal duration was not affected. In primary bovine chondrocytes, TGF-β-induced pSMAD2/3C lasted up to at least 3 h after stimulation, whereas pSMAD1/5C already disappeared after 3 h TGF-β stimulation. In both cell types, the length of the pSMAD signal was not affected by addition of IL-1β or OAS-cm (Figure 3B). Together, these data indicate that the inhibitory effect that we found on TGF-β signaling, was caused downstream of receptor-mediated SMAD activation. One such mechanism is through the induction of inhibitory SMAD7; however, in our experiments, *SMAD7* expression levels were not increased by inflammatory stimuli (Figure 3C). The exception was the 1 h stimulation with 5% OAS-cm, which did increase *SMAD7* expression in bovine chondrocytes but not in SW1353 cells.

### 2.4. IL-1β and OAS-cm Inhibit TGF-β via(de-)Phosphorylation of the SMAD2/3 Linker Region

Aside from C-terminal phosphorylation, SMAD proteins can also be post-translationally phosphorylated at serine and threonine residues within the linker region: SMAD2 at threonine (T) 220 and serines (S) 245, 250, 255 and SMAD3 at the corresponding T179, S204, S208 and S213 [17] (Figure 4A). TGF-β-induced transcriptional activity is regulated by SMAD linker modifications in several cell types [29,30,31,32]; therefore, we hypothesized that SMAD2/3 linker modifications are responsible for the effect of IL-1β and OAS-cm on TGF-β signaling in chondrocytes. We studied linker phosphorylation at those specific linker threonine and serine residues by Western blotting. SMAD2 and SMAD3 linker threonine and serine modifications were detectable within 1 h following IL-1β or OAS-cm stimulation (Figure 4B,C). Concentrations of 0.1, 1 and 10 ng/mL IL-1β induced phosphorylation of SMAD2L serines and SMAD3L S204 (*p* = 0.0256), but did not change pSMAD3L S208 (*p* = 0.1333) or S213 (*p* = 0.7633). Additionally, OAS-cm did induce pSMAD2L serines and pSMAD3L S204 (*p* = 0.0232), but not pSMAD3L S208 (*p* = 0.0973) and it significantly decreased the phosphorylation of SMAD3L S213 (*p* = 0.0047). The phosphorylation of SMAD2L T220 was not induced by IL-1β, but only by OAS-cm, whereas SMAD3L T179 was not regulated by either stimulus. These data suggest a role for especially serine linker modifications in regulating TGF-β signaling in chondrocytes.

To further explore the importance of SMAD linker modifications, we used five different SMAD3 variants, which cannot be phosphorylated at specific sites in the linker domain due to mutations from the linker serines to alanines and the linker threonine to a valine (Figure 5A). Equal over-expression of the different SMAD linker variants was checked with flow cytometry (Appendix C). In all conditions, over 90% of the cells were positively stained for FLAG and the geometric mean of this over-expression was not different, demonstrating that all SMAD3 variants were overexpressed equally to facilitate a fair comparison between conditions.

We pre-incubated 1 h with 0.1 ng/mL IL-1-β or 5% OAS-cm and then stimulated cells for 5 h with 0.5 ng/mL TGF-β after which luciferase signal was measured. Similarly to before, in the condition with normal SMAD3 variant, results indicated inhibition of SMAD2/3 transcriptional activity with 0.1 ng/mL IL-1-β or 5% OAS-cm. However, in chondrocytes over-expressing SMAD3 S204A or T179V mutant inhibition with IL-1-β was no longer statistically significant (Figure 5B). We compared the percentage inhibition of 0.5 ng/mL TGF-β with these inflammatory stimuli between the normal SMAD3 transduced cells and the conditions with the SMAD linker mutants (Figure 5C). When we overexpressed a SMAD mutant which could not be phosphorylated at the S204, we observed the trend that TGF-β signaling was less inhibited by IL-1β by an average of 32% (*p* = 0.10) in four separate experiments. We did not observe this with OAS-cm (*p* = 0.96). Remarkably, the inhibiting effect of both IL-1β and OAS-cm was significantly stronger when serine 213 could not be phosphorylated with 59% (*p* = 0.001) and 46% (*p* = 0.0003), respectively (Figure 5C). In Figure 5D, we summarized our findings regarding the effect of OA-related inflammation-induced dephosphorylation of SMAD3L S213 on SMAD2/3 transcriptional activity and of IL-1β-induced SMAD3L S204 modification.

## 3. Discussion

In this study, we showed that there is a link between OA-related inflammation and disturbed TGF-β signaling in chondrocytes. Our results indicate that IL-1β and OAS-cm can stimulate hypertrophy-like differentiation by decreasing TGF-β transcriptional activity. In addition, we demonstrate that the inhibition of TGF-β signaling was significantly enhanced when the SMAD3 linker phosphorylation on S213 cannot take place, while inhibition is possibly less pronounced when S204 cannot be phosphorylated. These observations indicate an important role for these modifications in regulating SMAD2/3 signaling in chondrocytes.

OA is a complex and multifactorial disease and it is recognized that both systemic and local inflammation disturb homeostasis of cartilage in the osteoarthritic joint [1,2]. The enhanced expression of IL-1β and its receptor (IL1R1) are found in chondrocytes and synovial membranes of OA patients [33,34], although its role in OA is still under debate [8,35,36,37,38]. OA can certainly not only be attributed solely to the effect of IL-1β and other pro-inflammatory cytokines contribute to OA pathogenesis [39,40,41,42,43,44]. For instance, IL-8, TNF-α and H_2_O_2_ also stimulate chondrocyte hypertrophy [45,46,47]. In this study, both IL-1β and patient-derived OAS-cm, containing an unknown mix of cytokines, chemokines and growth factors [48], were used as models of OA-related inflammation.

A crucial role for TGF-β in chondrocytes is controlling hypertrophy and blocking chondrocyte terminal differentiation through SMAD2/3 signaling [11,49]. In this study, we support this anti-hypertrophic effect of TGF-β, since it blocks *COL10A1* upregulation, the most evaluated hypertrophy marker in bovine cartilage explants. Importantly, we also showed that the pre-incubation of 0.1 ng/mL IL-1β or 0.5% OAS-cm before the addition of TGF-β clearly negated this inhibitory effect. These inflammatory stimuli also blocked SMAD2/3 transcriptional activity in three different human chondrocyte-like cell lines—G6, H11 and SW1353. This effect was quite strong and rapid, since only 1 h pre-incubation with 0.1 ng/mL IL-1β or 5% OAS-cm was sufficient to inhibit SMAD2/3 transcriptional activity for 47 and 64%, respectively, in SW1353 cells. Together, these results support the findings that OA-related inflammation blocks protective TGF-β signaling in chondrocytes. Previous studies reported similar interactions between pro-inflammatory stimuli and TGF-β signaling in chondrocytes [10,50]. For instance, Roman-Blas et al. reported that IL-1β treatment resulted in the suppression of the DNA-binding activity of SMAD3/4 and suppression of SMAD2/3 phosphorylation in chondrocytes [10] and Madej et al. showed that both IL-1β and OAS-cm impair the mechanical activation of SMAD2/3 signaling in bovine cartilage explants [50].

One way how inflammatory cytokines can modulate TGF-β induced pSMAD2/3 signaling is via a reduction in ALK5 or TGFBR2 receptor signaling [15,51]. In our experimental set-up, this is unlikely the explanation of the observed inflammation-induced inhibiting effect on SMAD2/3 transcriptional activity. Namely, our findings show that IL-1β or OAS-cm, in both primary bovine chondrocytes and SW1353 cells, did not affect C-terminal SMAD2/3 and SMAD1/5/8 phosphorylation. In support, no *ALK5* and *TGFBR2* mRNA downregulation was measured with IL-1β or OAS-cm stimulation. In SW1353 cells, *TGFBR2* was even induced (and not reduced) 6 h after IL-1β stimulation. Based on these results, we infer that the inhibitory effect, which we found on TGF-β signaling, is downstream of the receptor-mediated SMAD activation. Madej et al. also reported no effect of IL-1β or OAS-cm on their own on *ALK5* and *TGFBR2* receptor expression in bovine cartilage explants, while these inflammatory conditions partly suppressed the mechanically mediated SMAD2/3 signaling [50]. On the other hand, Baugé et al. showed that pro-inflammatory mediators such as IL-1β can reduce *TGFBR2* expression in human OA monolayer chondrocytes [15]. This might be due to the fact that they used OA chondrocytes for their study, which might react differently on cytokines than non-OA chondrocytes, which we used in our studies. Other than the modulation of receptor expression, IL-1β can increase the expression of inhibitory SMAD7 via NF-κB the activation in chondrocytes, which inhibits SMAD2/3 signaling [9]. However, in our study, short-time periods of 1 and 6 h with IL-1β did not result in increased *SMAD7* mRNA levels in primary bovine chondrocytes and SW1353 cells. Additionally, Roman-Blas et al. reported that SMAD7 is not involved in the suppression of TGF-β signaling induced by IL-1β [10]. Stimulation with OAS-cm for 1 h induced SMAD7 in bovine chondrocytes, but this effect disappeared 6 h after stimulation. This could possibly be explained by the TGF-β presence in OAS-cm, which also increased the expression of inhibitory *SMAD7* itself [27], since it was not shown for IL-1β.

Next, we investigated if IL-1β and OAS-cm interact with SMAD-dependent signaling through the modification of the SMAD2/3 linker region. Previous studies showed that the phosphorylation of the specific serine and threonine residues in the regulatory linker region control SMAD2/3 function. Mutations in the SMAD3 linker strongly enhanced TGF-β-induced responses in breast cancer cells and increased tumorigenesis in the liver [30,52]. SMAD2 linker phosphorylation elevated mRNA levels of glycosaminoglycan synthesizing enzymes in vascular smooth muscle cells [53] and also the phosphorylation of the SMAD2 linker mediates synthesis of extracellular matrix proteins, such as collagens and proteoglycans [31,32,54]. The phosphorylation and dephosphorylation of the serine and threonine residues in the linker domain is dependent on kinases (e.g., MAPK) and phosphatases (e.g., DUSP1) [17,22,23,55,56]. Particularly, these are also induced by OA-related inflammatory stimuli [57,58,59], which led us to hypothesize that inflammation-induced kinases or phosphatases also affect the SMAD linker region in chondrocytes. We reported earlier that IL-1β induces SMAD2L serine phosphorylation in stem cells [60]. In the current study, we also reported that in chondrocytes phosphorylation of the SMAD2L serines and SMAD3L S204 were observed within 1 h following IL-1β or OAS-cm stimulation. Notably, OAS-cm also significantly decreased pSMAD3L S213. pSMAD3L S208 and T179 were not regulated by IL-1β or OAS-cm, suggesting a less pronounced role of these linker modifications in blocking protective TGF-β effects in chondrocytes. In our study, we did not only show which SMAD linker residues were (de-)phosphorylated by inflammatory stimuli, but also examined whether these specific inflammation-induced linker modifications explain the observed inhibiting effects of inflammation on TGF-β signaling by using individual SMAD3 phospho-mutants. Most other studies make use of a SMAD2/3 EPSM mutant, which cannot be phosphorylated in the linker region on any phospho-site [29,61,62]. Using individual SMAD3 linker phospho-mutants, we investigated the effect of every single SMAD linker modification separately.

To further study the role of the inflammation-induced pSMAD3L S204, we made use of a SMAD3 mutant which could not be phosphorylated at the serine 204 site (S204A). The inhibition of the SMAD2/3 transcriptional activity was not significantly inhibited anymore with IL-1β when SMAD3 S204A was over-expressed, while this was the case when normal SMAD3 was over-expressed. However, the average effect of 32% less inhibition with IL-1β was not significant (*p* = 0.10) compared to the inhibiting effect in the condition using normal SMAD3. A high variation between samples could explain this non-significance. For this study, the unstable nature of the SBE-pNL1.2 luciferase construct was chosen for its high sensitivity and large detection window compared to other stable luciferases [63]. Since the direction of the effect observed with the SMAD3 S204 mutant was constant across four separate experiments, we carefully propose that SMAD3L S204 phosphorylation mediates the effect of IL-1β on SMAD2/3 signaling (summarized in Figure 5D). Linker modifications are able to regulate the nuclear localization of the SMADs and this could be the possible explanation why in our study SMAD3 S204 was essential for the blocking effect of IL-1β on SMAD2/3 transcriptional activity. Kretzschmar et al. reported that in a mouse mammary epithelial cell line, Ras-activated ERK-induced pSMAD3 S204 resulted in cytoplasmic retention and the consequent repression of canonical TGF-β signaling [61]. Additionally, in epithelial cells, excessive Ras signaling demonstrated lower pSMAD3C tumor suppression [64,65]. A similar process could take place in chondrocytes and explain our results. On the other hand, contradictory findings were reported in different cell types. For instance, it was reported that in fibroblasts and mesangial cells, ERK-induced pSMAD3L S204 enhanced SMAD3-mediated COL1A2 promotor activity [66] and glycogen synthase kinase 3 (GSK3)-induced pSMAD3L S204 was strengthening SMAD3 transcriptional activity by enhancing its affinity to CREB-binding protein [23]. SMAD signaling could also be regulated via binding to ubiquitin ligases, such as Smurf2 or NEDD4L, resulting in SMAD degradation [67,68], but SMAD3 S204 phosphorylation has not been reported to regulate SMAD3 stability [67,69,70]. Another explanation could be the binding of pSMAD3L S204 to the phosphatase PPM1A/PP2Cα, which is known to dephosphorylate the SMAD2/3 C-terminus, and thereby regulate TGF-β signaling [71,72]. However, such interaction has not yet been investigated and further research into why SMAD3 S204 phosphorylation is essential for the inhibitory effect of IL-1β in chondrocytes is required.

The effect of OAS-cm on TGF-β signaling was not inhibited using the SMAD3 S204A mutant, while OAS-cm stimulation induced the phosphorylation of S204 on Western blot. This suggests that the inhibition by OAS-cm is regulated differently than with IL-1β, and S204 phosphorylation is not required to allow the OAS-cm-induced inhibition of SMAD2/3 transcriptional activity. OAS-cm is a mixture of cytokines which all have diverse roles on the SMAD3 linker and follow different kinetics of (de-)phosphorylation. The functional outcome of the SMAD2/3 linker phosphorylation for SMAD2/3 transcriptional activity depends on the combination of phosphorylation sites in linker and C-terminal regions, which brings some levels of complexity. This could explain why we did find S204 phosphorylation with OAS-cm on Western blot when looking at it as a single object, but did not observe an effect on the OAS-cm-induced inhibition of SMAD2/3 transcriptional activity when this phosphorylation could not occur anymore.

Interestingly, Browne et al. found an opposing role for SMAD3L S213 compared to S204 phosphorylation on COL1A2 promotor binding [66]. This is consistent with our study where we also showed contradictory results for SMAD3L S204 and S213 phosphorylation. Namely, the inhibiting effect of both IL-1β or OAS-cm was significantly enhanced when SMAD3L S213 could not be phosphorylated with 59% and 46%, respectively. This suggests that the phosphorylation of SMAD3L S213 protected against the inhibiting effect of IL-1β and OAS-cm on transcriptional SMAD2/3 signaling (summarized in Figure 5D). In literature, several lines of evidence report that SMAD3L S213 phosphorylation, induced by the Ras/JNK pathway, results in the transport of SMAD3 to the nucleus [16,64]. This would suggest that with the dephosphorylation of the S213 site SMAD2/3 remains in the cytoplasm and thereby prevents transcriptional activity. We found that OAS-cm significantly decreased the phosphorylation of SMAD3L S213, and thereby OAS-cm contributed itself to the inhibition on SMAD2/3 transcriptional activity. We reported earlier that combined IL-1β and TGF-β treatment in stem cells resulted in more linker-modified SMAD2 in the cytoplasm and less nuclear pSMAD2C [60]. Other studies showed that IL-1β-induced TAK activity resulted in cytoplasmic retention of the SMADs [73,74]. In future studies, the effect of linker modifications on the cellular localization of the SMAD complexes should, therefore, be examined.

The phosphorylation of SMAD3 S213 is protective against OA-related inflammation in chondrocytes. As a therapeutic strategy, it would be possible to activate kinases that are known to phosphorylate S213 in chondrocytes. Ras/JNK, CDK2, CDK4, SKI and integrin all have been reported to enhance pSMAD3L S213 phosphorylation in different cell types [70,75,76,77]. Another option is to inhibit phosphatases which catalyze the removal of phosphate groups. Small C-terminal domain phosphatases (SCPs) are known to dephosphorylate pSMAD3L S213 in the nucleus, resulting in the dissociation from SMAD4 and the export of SMAD3 to the cytoplasm [56,71,78,79]. SCP, therefore, could be an interesting therapeutic target. Blocking it could enhance the protective TGF-β signaling through the inhibition of dephosphorylation of the SMAD3 S213 linker phosphor-site by inflammation, resulting eventually in more SMAD3 in the nucleus. The identification of an inhibitor for SCP1 is ongoing [80,81]. For future in vivo studies, one must careful use these SCP inhibitors, since the phosphorylation of SMAD3 S213 results in cell-type specific effects. Namely, the nuclear retention of pSMAD3L S213 is reported to enhance pro-oncogenic signaling in cancer cells, by facilitating mitogenic signaling via the upregulation of the transcription factor c-Myc [64,70,75,76,77,82]. This shows that the phosphorylation of SMAD3L S213 can be malicious in cancer cells, while in chondrocytes S213 phosphorylation seems to be protective against OA-related inflammation. The discrepancy between these observations in different cell-types warns us to extract these results and more cell-specific research on the function of SMAD linker modifications is needed.

It is a limitation of this study that we were not able to transfect primary chondrocyte explants with the SMAD3 linker variants. Therefore, we could not test if the linker region was important in regulating the hypertrophic differentiation of chondrocytes in our hypertrophy model. Future studies are needed for the identification of interacting proteins of the SMAD2/3 linker domain. For example, it would be of great interest to study the difference in the interaction of normal SMAD3 versus SMAD3 S204 or S213 linker mutants with RUNX2/3 and MEF2C, which are important transcription factors in driving chondrocyte hypertrophy [83,84].

In conclusion, the SMAD2/3 linker region is critical for the regulation of TGF-β signaling. The relevance of SMAD linker modifications in fibrosis, cancer and cardiovascular disease was described earlier but not in joint diseases. In this study, also the relevance for chondrocytes was established. Joint inflammation during OA development will result in kinase and phosphatase activation that could (de-)phosphorylate the SMAD linker region independent of the C-terminal phosphorylation, including the S204 and S213 site [85]. We showed that the (de)-phosphorylation of these linker sites led to a disturbance of the TGF-β signaling pathway in cartilage, which is of great importance for chondrocyte homeostasis maintenance. An additional investigation in chondrocytes is needed to identify the specific kinases and phosphatases for the individual SMAD linker phospho-sites, the impact of these modifications on the cellular location of the SMAD-complexes and the functional consequences for the cartilage. Inhibition studies of relevant kinases and phosphates may result in new therapeutic targets for OA.

## 4. Materials and Methods

### 4.1. Primary Cell Culture

Articular cartilage was obtained from metacarpophalangeal joints (MCP) of skeletally mature cows (>3 years old) post mortem. Full cartilage thickness explants were isolated with 3 mm diameter biopsy punches (Kai Medical Seki, Japan) and randomly distributed over the different conditions (two times four explants per cow per condition). Explants were equilibrated overnight before start of experiments in DMEM/F12 medium, supplemented with 100 mg/l sodium pyruvate, 100 U/mL penicillin and 100 µg/mL streptomycin at 37 °C and 5% CO_2_. To obtain chondrocytes to culture in monolayer, cartilage slices were digested overnight with 1.5 mg/mL collagenase B (Roche Diagnostics, Basel, Switzerland) in DMEM/F12 at 37 °C. The next day, chondrocyte suspension was spun down at 300× *g* for 10 min, washed three times using saline and resuspended in DMEM/F12 containing 10% fetal calf serum (FCS), 100 mg/L sodium pyruvate, 100 U/mL penicillin and 100 µg/mL streptomycin (complete DMEM/F12). Chondrocytes were plated at a density of 8 × 10^4^ cells/cm^2^ and cultured for 1 week at 37 °C and 5% CO_2_ to form a monolayer. Medium was refreshed every three days. Before start of experiments, chondrocytes were serum-starved (0% FCS) overnight. Each experiment was conducted three times in multiple donors and conditions were always tested in technical duplicate.

### 4.2. Chondrocyte Cell Line Culture

SW1353 human chondrosarcoma cells were cultured in complete DMEM/F12 at 37 °C and 5% CO_2_. For experiments, cells were plated at a density of 3 × 10^4^ cells/cm^2^. Human G6 and H11 adult articular chondrocytes were derived from femoral head cartilage of an anonymous donor, transduced with a temperature-dependent SV40 large oncogene, resulting in proliferation at 32 °C, but not at 37 °C [86]. G6 and H11 chondrocytes were cultured at 32 °C with complete DMEM/F12 except with 5% FCS. For experiments G6 and H11, cells were plated at a density of 8 × 10^4^ cells/cm^2^. Chondrocytes were serum-starved overnight in DMEM/F12 medium supplemented with 0.1% FCS (SW1353) or 0.5% FCS (G6 and H11) before start of experiments.

### 4.3. Chondrocyte Stimulation

Chondrocytes were stimulated with recombinant human (rh) TGF-β1 (BioLegend, San Diego, CA, USA), rhIL-1β (R&D Systems, Minneapolis, MN, USA), OA synovium-conditioned medium (OAS-cm), or a combination of these mediators, for time periods and dosages indicated in Figure legends. OAS-cm was obtained by culturing synovium from OA patients for 24 h, whereafter debris was removed by centrifugation at 300× *g* and medium was stored in aliquots at −20 °C until further use [48]. To inhibit ALK5 kinase activity, 5 µM SB-505124 (Sigma-Aldrich, Burlington, MA, USA) was used, dissolved in dimethyl sulfoxide (DMSO).

### 4.4. Plasmid DNA, Adenoviral Production and Transduction

To study SMAD2/3 transcriptional activity, a luciferase reporter assay (SBE-pNL1.2) was produced, where a SMAD binding element (SBE) (three times AGTATGTCTAGACTGA) with spacer (CTCGAGGATATCAAGATCTGGCCTCGGCGGCCTAGATGAGACACT) and minimal promotor (AGAGGGTATATAATGGAAGCTCGACTTCCAG) (GeneCust, Boynes, France) was cloned into a NanoLuc luciferase with a protein destabilization domain (pNL1.2) (Promega, Madison, WI, USA) [63]. Sequences were verified by Sanger sequencing. Knock-out of SMAD2, SMAD3 or SMAD4 prevented luciferase induction with TGF-β, suggesting the reporter assay is SMAD2/3-dependent (Appendix B). Plasmid transduction was optimized for the different cell lines by analysis of fluorescent (GFP) protein expressing cells with FACS. G6 and H11 chondrocytes were seeded in a cell density of 8 × 10^4^ cells/cm^2^ and transfected with Lipofectamine 2000 Transfection Reagent (Invitrogen, Waltham, MA, USA) according to manufacturer’s protocol. SW1353 were seeded in a density of 2.6 × 10^4^ cells/cm^2^ and transfected with FuGENE6 Transfection Reagent (Promega, Madison, WI, USA) according to manufacturer’s protocol. SMAD3 linker mutant expression plasmids, containing an N-terminal FLAG-tag, were bought from Addgene (Watertown, MA, USA) (SMAD3, #14052; SMAD3 T179V, #26997; SMAD3 S204A, #27114; SMAD3 S208A, #27115; SMAD3 S213A, #27116; SMAD3 EPSM, #14963). All SMAD inserts were directionally cloned into the adenoviral vector pShuttle and verified by Sanger sequencing. Adenovirus was produced with the AdEasy Adenoviral Vector System (Agilent, Santa Clara, CA, USA) in the N52E6 adenovirus producer cell line. SW1353, already transfected with SBE-pNL1.2, was transduced with adenovirus of the different SMAD3 linker mutants. To compare equal over-expression of the different mutants, flow cytometry was used to quantify FLAG-tag expression with PE anti-FLAG tag Antibody (clone L5, Biolegend, San Diego, CA, USA) using a Gallios flow cytometry analyzer and analyzed using Kaluza software version 2.1 (both from Beckman Coulter, Brea, CA, USA).

### 4.5. SMAD-Luciferase Transcriptional Reporter Assay

After transfection with 1.0 µg SBE-pNL1.2 per 100,000 cells, cells were detached by trypsinization and seeded in white polystyrene 96-well plates at a density of 3 × 10^4^ cells/cm^2^ for the SW1353 and 8 × 10^4^ cells/cm^2^ for the G6 and H11 chondrocytes. Cells were serum-starved overnight, 1 h pre-incubated with DMEM/F12 (control), rhIL-1β (R&D Systems, Minneapolis, MN, USA) or OAS-cm and then stimulated with rhTGF-β1 (Biolegend, San Diego, CA, USA) for 5 h. Cells were lysed 5 h post-stimulation using 30 µL ultra-pure water. An equal amount of Nano-Glo luciferase reagent (Promega, Madison, WI, USA) was added and luminescence was determined at 470–480 nm (Clariostar, BMG Labtech, Ortenberg, Germany). Each condition was performed in quadruple and the mean per experiment was depicted.

### 4.6. Protein Isolation and Western Blot

Chondrocytes were lysed in lysis buffer (Cell Signaling, Danvers, MA, USA) containing complete protease inhibitor cocktail (Roche Diagnostics, Basel, Switzerland). Samples were sonicated on ice, using a Bioruptor (Diagenode, Liege, Belgium; 10 cycles of 30 s sonication and 30 s rest). Protein concentration was determined with a BCA-assay (Thermo Scientific, Waltham, MA, USA) and normalized. Reducing Laemmli Sample buffer (2% SDS, 10% glycerol, 100 mM Tris HCl, pH 6.8, 100 mM DTT and Bromophenol bleu) were added and samples were boiled at 95 °C for 5 min. Protein samples were separated on a 10% bis-acrylamide SDS-PAGE gel and transferred to 0.45 µm pore nitrocellulose membrane using wet transfer (Towbin buffer, 2 h, 275 mA at 4 °C). Non-specific antibody binding was blocked for 1 h with 5% non-fat dry milk (Friesland Campina, Amersfoort, The Netherlands) or 5% BSA in TBS-T (15 mM Tris-HCl, pH 7.4, 0.1% Tween). Cells were incubated overnight at 4 °C with primary antibodies directed against pSMAD2/3C-Ser463/467 (1:1000, CST 3101), pSMAD1/5C-Ser426/428 (1:1000, CST 9511), pSMAD2L-Ser245/250/255 (1:1000, CST 3104), pSMAD2L-Thr220 (1:1000, NBP 1-004982), pSMAD3L-Ser204 (1:1000, Abcam 63402), pSMAD3L-Ser208 (1:1000, Abcam 138659), pSMAD3L-Ser213 (1:1000, Abcam 63403), pSMAD3L-Thr179 (1:1000, Abcam 74062) or anti-FLAG (1:10,000, Sigma-Aldrich 3165). Afterwards, membranes were incubated with polyclonal Goat anti-Rabbit or Rabbit anti-Mouse coupled to horseradish peroxidase (1:1500, Dako) for 1 h at RT. Signal was detected using enhanced chemiluminescence (ECL) prime kit (GE Healthcare, Chicago, IL, USA) on an ImageQuant LAS4000 (Leica, Wetzlar, Germany). As loading control, GAPDH (1:10,000, Sigma-Aldrich 1403850) was used and ImageJ (Fiji 1.51n) was used for quantification.

### 4.7. RNA Isolation and Quantitative Real-Time PCR

Cartilage explants were homogenized using a micro-dismembrator (B. Braun, Oss, The Netherlands) for 1 min at 1500 rpm. Subsequently, total messenger RNA (mRNA) was isolated using RNeasy Fibrous tissues kit (Qiagen, Hilden, Germany) according to manufacturer’s protocol. From cell lines and primary chondrocytes cultured in monolayer, mRNA was isolated using 500 µL TRIzol (Sigma-Aldrich, Burlington, MA, USA), according to manufacturer’s protocol. After isolation, a maximum of 1 µg of mRNA was treated with 1 µL DNAse (Life Technologies, Carlsbad, CA, USA) for 15 min at room temperature to remove possible genomic DNA, followed by 10 min inactivation by incubation at 65 °C with 1 µL 25 mM EDTA (Life Technologies). mRNA was reverse transcribed to complementary DNA using 1.9 µL ultrapure water, 2.4 µL 10x DNAse buffer, 2.0 µL 0.1 M dithiothreitol, 0.8 µL 25 mM dNTP, 0.4 µg oligo dT primer, 200 U M-MLV reverse transcriptase (all Life Technologies, Carlsbad, CA, USA) and 0.5 µL 40 U/mL RNAsin (Promega, Madison, WI, USA) and incubated for 5 min at 25 °C, 60 min at 39 °C, and 5 min at 95 °C using a thermocycler. Gene expression was measured using SYBR Green Master Mix (Applied Biosystems, Waltham, MA, USA) and 0.25 mM primers (Biolegio, Nijmegen, the Neterlands) (Table 1) with a StepOnePlus real-time PCR system (Applied Biosystems, Waltham, MA, USA). The amplification protocol was 10 min at 95 °C, followed by 40 cycles of 15 s at 95 °C and 1 min at 60 °C. Melting curves were analyzed to confirm product specificity. To calculate the relative gene expression (−ΔCt), the average of three reference genes was used: *bGAPDH*, *bRPL22* and *bRPS14* for bovine chondrocytes or *hGAPDH*, *hRPL22* and *hRPS27A* for human chondrocyte cell lines.

### 4.8. Statistical Analysis

Quantitative data of gene expression analysis were expressed as column scatter graphs and displayed mean values of a technical duplicate sample per donor (primary chondrocytes) or separate experiments (SW1353 cells) with corresponding 95% confidence interval (CI) or standard deviations (SD) (see Figure legends). For SBE-pNL1.2 transcriptional assays, conditions were investigated in quadruplo and expressed as mean per experiment with corresponding 95% confidence interval (CI). Area under the curve (AUC) was calculated for three separate experiments. Differences were tested using displayed means with analysis of variance (ANOVA) followed by Dunnett’s or Bonferroni’s post-test to take multiple comparisons into account (see Figure legends). Differences in pSMAD2 and pSMAD3L protein were tested using an unpaired two-tailed *t*-test and displayed as mean ± SD. Statistical differences were considered as significant if the *p*-value was below 0.05. All analyses were performed using Graph Pad Prism version 7.0 (GraphPad Software, San Diego, CA, USA).

## Figures and Tables

**Figure 1 ijms-22-08124-f001:**
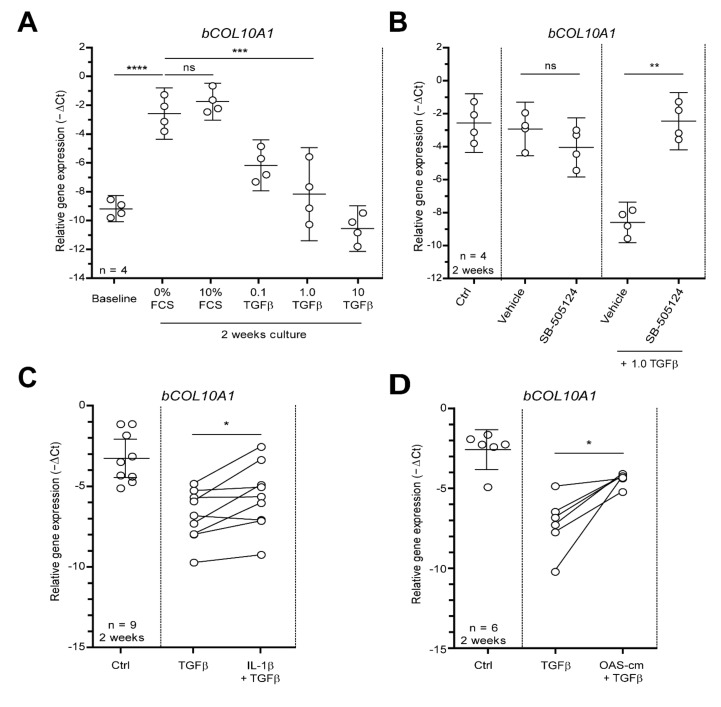
OA-related inflammation blocks anti-hypertrophic function of TGF-β in bovine cartilage explants. To induce hypertrophy-like differentiation, bovine cartilage tissue explants were cultured ex vivo for 2 weeks and medium was replaced every 3rd day. (**A**) Culturing cartilage explants for 2 weeks with or without FCS spontaneously induced hypertrophy-like differentiation, as measured by relative collagen type 10 (*COL10A1*) mRNA expression using qPCR. To study the anti-hypertrophic role of TGF-β, the effect of different concentrations of rhTGF-β1 (0.1, 1 and 10 ng/mL) on *COL10A1* mRNA expression was measured (in medium without FCS). (**B**) Co-incubation with 5 µM ALK-5 kinase activity inhibitor SB-505124 fully blocked TGF-β (1 ng/mL) effects on *COL10A1* mRNA expression compared with vehicle (DMSO). (**C**,**D**) To study the interaction between TGF-β and inflammatory mediators in this model of hypertrophy, explants were exposed to 0.1 ng/mL TGF-β with 1 h pre-incubation of 0.1 ng/mL IL-1β (**C**) or 0.5% OAS-cm (**D**). Data are plotted as mean ± 95% CI with each dot representing the average of 2 replicates of 4 explants in one cow. Statistical analysis was performed using a repeated measures one-way analysis of variance with Bonferroni’s post hoc test (A + B) or a two-tailed Student’s paired *t*-test (C + D): ns non-significant *p* > 0.05; * *p* ≤ 0.05; ** *p* ≤ 0.01; *** *p* ≤ 0.001; **** *p* < 0.001.

**Figure 2 ijms-22-08124-f002:**
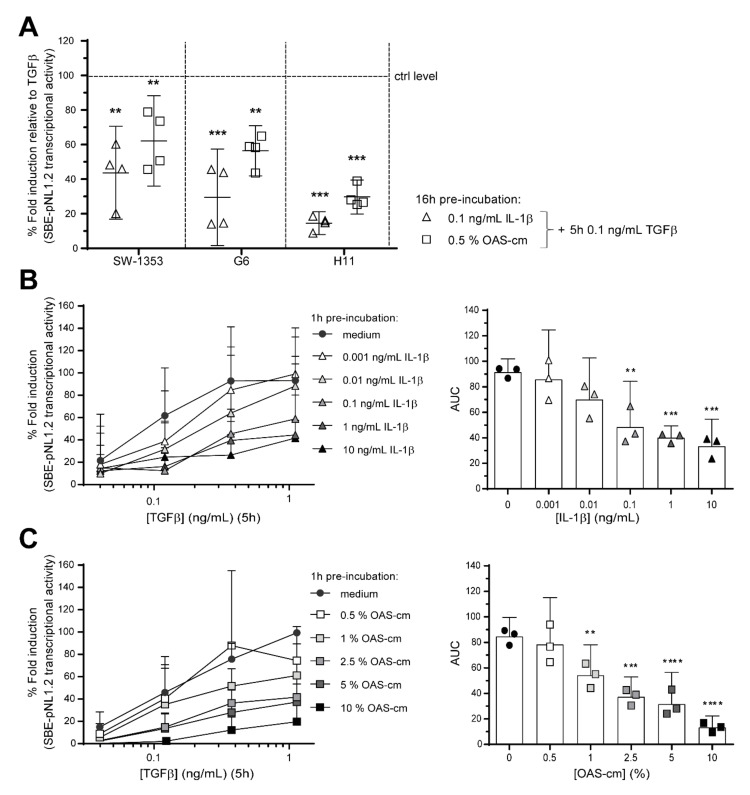
OA-related inflammation inhibits TGF-β transcriptional activity in particular chondrocytes. To study interaction between OA-related inflammation and functional TGF-β signaling, three chondrocyte-like cell lines (SW1353, G6 and H11) were transfected with a SMAD2/3 transcriptional reporter construct (SBE-pNL1.2). (**A**) After transfection, cells were re-plated, pre-incubated overnight (16 h) with medium, 0.1 ng/mL IL-1β or 0.5% OAS-cm and, thereafter, stimulated for 5 h with 0.1 ng/mL TGF-β. Luciferase activity was measured relative to experimental condition stimulated with TGF-β, as set at 100% (ctrl level). Data represent mean ± 95% CI of four independent experiments performed in quadruple. (**B**,**C**) In SW1353 cells, this was investigated further, but now with 1 h pre-incubation with a concentration series of (**B**) IL-1β (0.001–10 ng/mL) or (**C**) OAS-cm (0.5–10%) before stimulation with increasing concentrations of TGF-β for 5 h. Data represent mean ± 95% CI of three independent experiments performed in quadruple. Per experiment the area under the curve (AUC) was calculated and displayed. Statistical analysis was performed using a one-way ANOVA with Dunnett’s multiple comparison test comparing the mean to the mean of the condition stimulated with solely TGF-β: ** *p* ≤ 0.01; *** *p* ≤ 0.001; **** *p* < 0.001.

**Figure 3 ijms-22-08124-f003:**
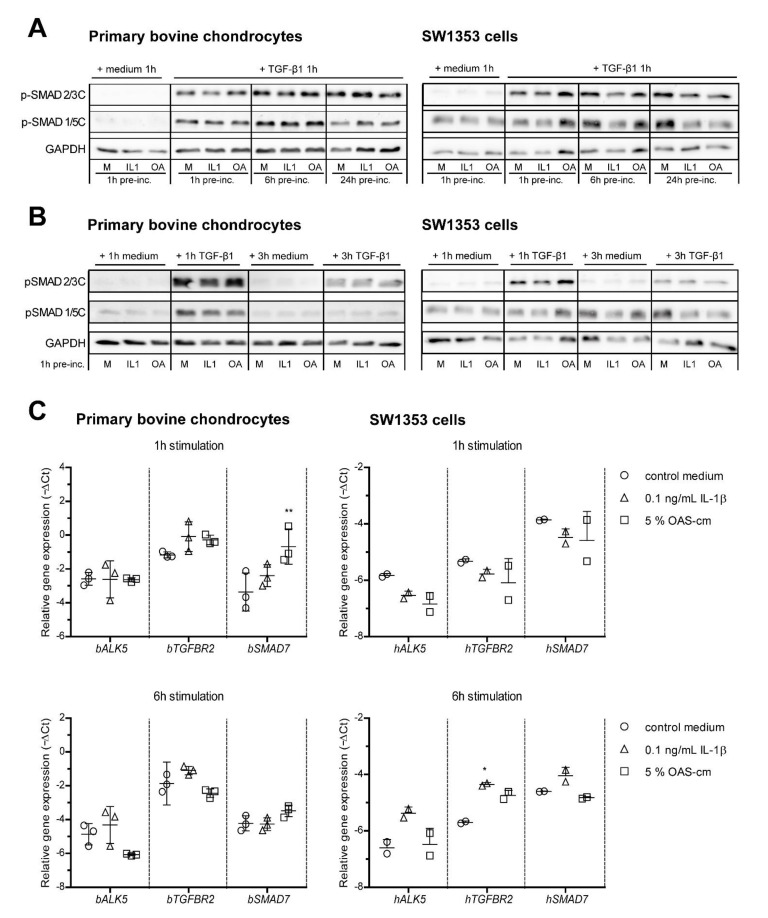
IL-1β and OAS-cm do not alter C-terminal phosphorylation of SMAD transcription factors and do not regulate receptor level expression. In search of an explanation for the inhibition of OA-related inflammatory factors on functional TGF-β signaling, we investigated if IL-1β and OAS-cm influence C-terminal SMAD2/3 and SMAD1/5 phosphorylation in bovine chondrocytes cultured in monolayer and in SW1353 chondrosarcoma cells using Western blot (**A**,**B**). Pre-incubation for different time periods (1, 6 and 24 h) with 0.1 ng/mL IL-1β (IL1) or 2.5% OAS-cm (OA) did not alter p-SMADC activation with 1 ng/mL TGF-β (**A**) and also signal duration was not affected (**B**). GAPDH was included as loading control. (**C**) Relative gene expression of TGF-β receptors ALK5 and TGFBR2 and inhibitory SMAD7 in bovine chondrocytes and SW1353 cells 1 h and 6 h after stimulation with medium supplemented with 0.1 ng/mL IL-1β or 5% OAS-cm. Data are plotted as mean ± SD with each dot representing the average of 2 replicates of 4 explants in one cow (*n* = 3), or in case of the SW1353 cells of two independent experiments performed in duplicate. Statistical analysis was performed using a one-way ANOVA with Bonferroni’s post hoc test: * *p* ≤ 0.05; ** *p* ≤ 0.01.

**Figure 4 ijms-22-08124-f004:**
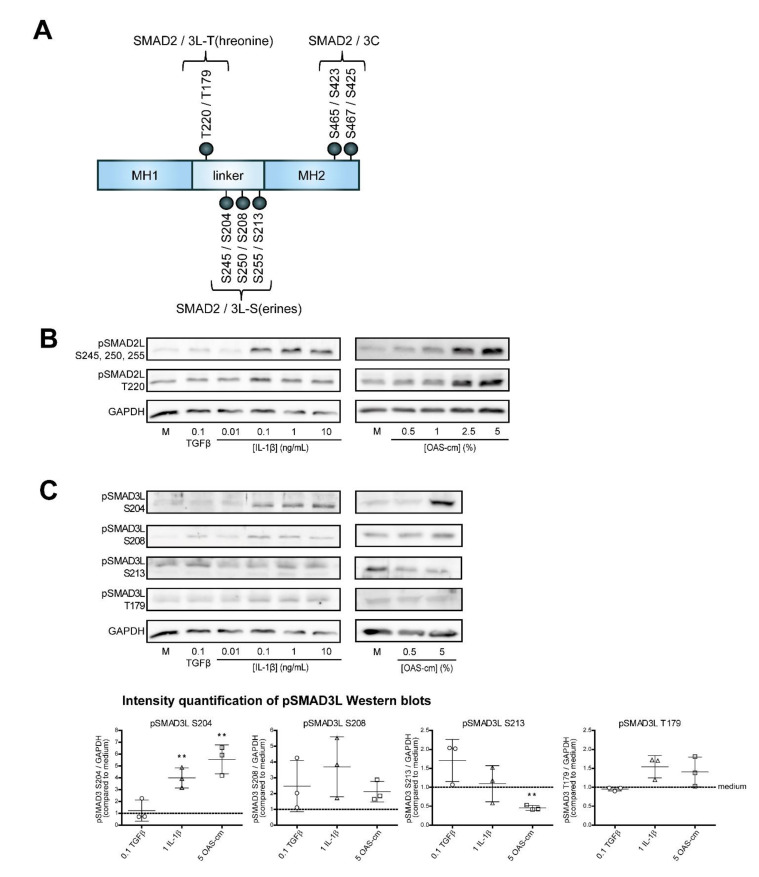
OA-related inflammation (de)phosphorylates the SMAD 2/3 linker region. SMAD proteins can also be post-translationally phosphorylated at serine and threonine residues within their linker (L) region: SMAD2 at threonine (T) 220 and serines (S) 245, 250, 255 and SMAD3 at the corresponding T179, S204, S208 and S213. (**A**) Schematic illustration of the SMAD2 and SMAD3 proteins and their phospho-epitopes in the linker (L) region and C-terminus. (**B**,**C**) Chondrocytes were treated for 1 h with 0.1 ng/mL TGF-β or with different concentrations of IL-1β (0.01–10 ng/mL) or OAS-cm (0.5–5%) and subsequently phosphorylation at the different phospho-sites in the linker region of SMAD2 (**B**) and SMAD3 (**C**) were visualized on Western blot. Quantification of the Western blots was performed with ImageJ. Data are presented as dot plots with mean ± SD, with each dot representing one donor, *n* = 3. GAPDH was used as loading control. Statistics were performed using two-tailed Student’s paired *t*-test: ** *p* ≤ 0.01.

**Figure 5 ijms-22-08124-f005:**
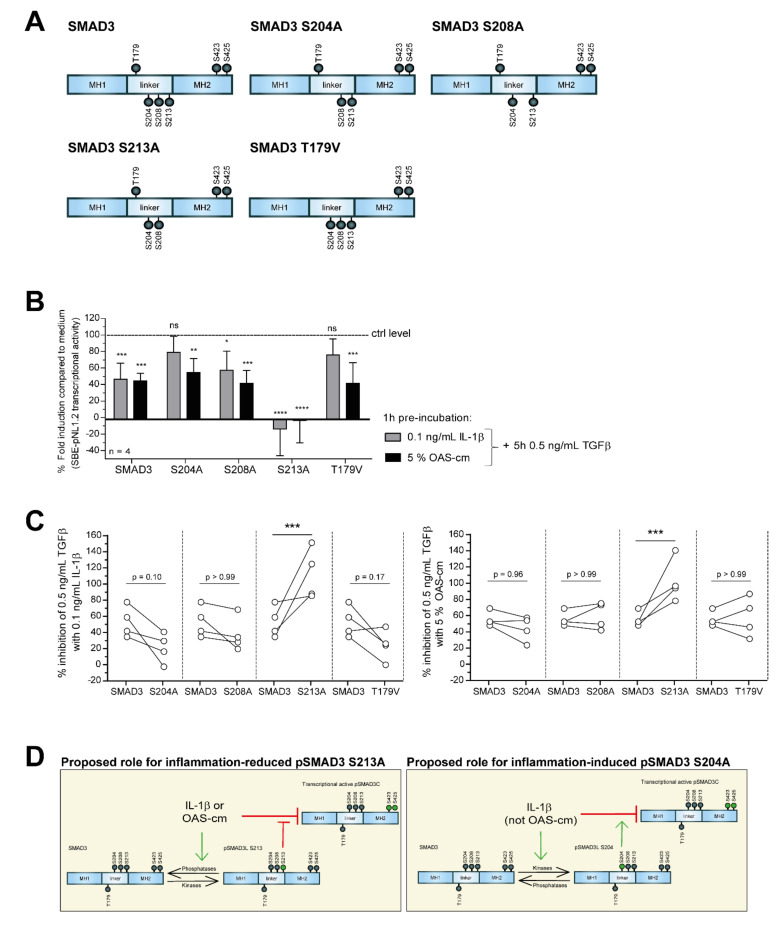
When the SMAD3 linker could not be modified, TGFβ transcriptional activity was differently regulated by OA-related inflammation. To study the role of the SMAD3 linker phospho-sites in regulating TGF-β signaling, we made use of individual SMAD3 linker variants. (**A**) Schematic illustration of the different SMAD3 variants which cannot be phosphorylated on specific sites in their linker domain due to mutations from the serines (S) to alanines (A) (S204A, S208A, S213A) and the threonine (T) to a valine (V) (T179V). (**B**,**C**) SW1353 cells were transfected with the SBE-pNL1.2 construct, re-plated afterwards and transduced with the different SMAD variants. After a 48 h transduction and after overnight serum-starvation, we pre-incubated 1 h with 0.1 ng/mL IL-1β or 5% OAS-cm and then stimulated with 0.5 ng/mL TGF-β, after which luciferase signal was measured. (**B**) Percentage fold induction compared to medium was depicted, relative to experimental condition stimulated with 0.5 ng/mL TGF-β set at 100% (ctrl level). Data represent mean ± SD of four independent experiments performed in quadruple. (**C**) The percentage inhibition of 0.5 ng/mL TGF-β with 0.1 ng/mL IL-1β or with 5% OAS-cm was calculated and compared between the normal SMAD3 transduced cells and the conditions transduced with the SMAD linker mutants. Every dot represents one independent experiment performed in quadruple. Statistical analysis was performed using a one-way ANOVA with Dunnett’s multiple comparison test. ns— non-significant; * *p* ≤ 0.05; ** *p* ≤ 0.01; *** *p* ≤ 0.001; **** *p* < 0.001. (**D**) Summarized findings regarding the effect of inflammation-induced dephosphorylation of SMAD3L S213 on SMAD2/3 transcriptional activity and the hypothetical role of IL-1β-induced S204 linker modification. Green arrows represent activation and red “T” shapes represent inhibition.

**Table 1 ijms-22-08124-t001:** Primer sequences as used in this study.

*Gene and Species*	Forward Sequence (5′ → 3′)	Reverse Sequence (5′ → 3′)
*bGAPDH*	CACCCACGGCAAGTTCAAC	TCTCGCTCCTGGAAGATGGT
*bRPS14*	CATCACTGCCCTCCACATCA	TTCCAATCCGCCCAATCTTCA
*bRPL22*	GTTCGCTCACCTCCCTTTCTG	GCAGCATCCATGATTCCATCT
*bCOL10A1*	CCATCCAACACCAAGACACAGT	TGCTCTCCTCTCAGTGATACACCTT
*bMMP3*	AAACTCACCTCACGTACAGAATTG	TCCCAGACCGTCAGAGCTTT
*bSMAD7*	GGGCTTTCAGATTCCCAACTT	CTCCCAGTATGCCACCACG
*bTGFBR2*	GGCTGTCTGGAGGAAGAATGA	GTCTCTCCGGACCCCTTTCT
*bALK5*	CAGGACCACTGCAATAAAATAGAACTT	TGCCAGTTCAACAGGACCAA
*hGAPDH*	ATCTTCTTTTGCGTCGCCAG	TTCCCCATGGTGTCTGAGC
*hRPL22*	TCGCTCACCTCCCTTTCTAA	TCACGGTGATCTTGCTCTTG
*hRPS27A*	TGGCTGTCCTGAAATATTATAAGGT	CCCCAGCACCACATTCATCA
*hSMAD7*	CCTTAGCCGACTCTGCGAACTA	CCAGATAATTCGTTCCCCCTGT
*hTGFBR2*	CTGGTGCTCTGGGAAATGACA	TCGCCCTCGATCTCTCAACA
*hALK5*	CGACGGCGTTACAGTGTTTCT	CCCATCTGTCACACAAGTAAA

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
