# Peer review of "Osteoarthritis-Related Inflammation Blocks TGF-β’s Protective Effect on Chondrocyte Hypertrophy via (de)Phosphorylation of the SMAD2/3 Linker Region"

_ijms, 2021, doi:10.3390/ijms22158124_

Round 1
Reviewer 1 Report
very usefull and interested paper
Beautiful works
Authors investigated whether OA-related inflammation dysregulates TGF-β responses in chondrocytes via inflammation and then SMAD2/3 protein linker-modifications.
They demonstrates that the inhibition of TGF-β signaling is significantly enhanced when SMAD3 linker phosphorylation on S213 cannot take place, while inhibition is possibly less pronounced when S204 cannot be phosphorylated. These observations demonstrate an important role for these modifications in regulating SMAD2/3 signaling in chondrocytes.
The phosphorylation of SMAD3 S213 is protective against OA-related inflammation in chondrocytes. As a therapeutic strategy it would be possible to activate kinases that are known to phosphorylate S213 in chondrocytes (or to inhibit phosphatases).
So the powerful interest in these demonstration is that when modulates dephosphorylation of the SMAD3 S213 linker phosphor-site we modulates protective TGF-β signaling.
In conclusion, the SMAD2/3 linker region is critical for regulation of TGF-β signaling in joint OA.
Author Response
Dear reviewer,
We would like to thank you for taking the time and effort to read and review our research paper, and thank you for your positive compliments. We were glad to hear you appreciate our work and see it as very useful and interesting. We look forward to follow up on this research and indeed inhibit/activate related kinases and phosphatases to target disturbed TGF-β signaling in joint osteoarthritis.
Yours Faithfully,
The authors
Reviewer 2 Report
This study demonstrates that OA-synovium conditioned medium and IL-1β blocks TGF-β’s protective effect on chondrocyte hypertrophy via phosphorylation or dephosphorylation of SMAD2/3 linker region. This study is very interesting, but I have a couple of minor comments.
- There is a difference of TGF-β-induced phosphorylation of SMAD2/3 linker region by treatment with between OAS-CM and IL-1βonly. The authors describe due to that OAS-CM contains as unknown mix of cytokines, chemokines and growth factors. However, it is well-known that TNF-α has an important role in OA-induced inflammation. Does TNF-α also block the effect of TGF-β?
- This study showed that OAS-CM strongly induces phosphorylation of SMAD3 S204 and dephosphorylation of SMAD3 S213. However, in Figure 5B, SMAD3 transfection activity was significantly inhibited by OAS-CM in chondrocytes transfected with which form of SMAD3 mutant. These results are confusing data. The authors should explain these results.
- In Fig. 4C, the bands of pSMAD3 S204 by IL-1β are very faint. I think that intensity of these bands does not consist with quantification data. The author should show the clear bands.
- The author indicate that although OAS-CM and IL-1β inhibit TGF-β signaling, these factors have no effect on phosphorylation of C-terminal SMAD2/3, which play an important role in binding to SMAD4. When SMAD2/3 linker region is (de)phosphorylated by OAS-CM and IL-1β, how is TGF-β signaling inhibited?
Author Response
Dear reviewer,
We would like to thank you for taking the time and effort to read and review our research paper. We were glad to hear you appreciate our work and see it as very interesting. We have tried to address your comments, included as PDF file.
Hopefully we have been able to answer your comments to your satisfaction.
Yours Faithfully,
The authors
